# The Global COVID-19 Pandemic Experience: Innovation Through Environmental Assessment and Seropositivity Surveillance

**DOI:** 10.3390/ijerph22071145

**Published:** 2025-07-18

**Authors:** Robert M. Park

**Affiliations:** National Institute for Occupational Safety and Health, Cincinnati, OH 45230, USA; rhpark9@gmail.com

**Keywords:** exposure–response, infection fatality ratio, natural immunity, seropositivity, vaccination, virus exposure, zero-COVID

## Abstract

Objectives: To confirm a conjecture from year 2020 of the SARS-CoV-2 (COVID-19) pandemic suggesting policy alternatives to substantially reduce mortality burden. Methods: Data from a global COVID-19 database comparing different countries on cumulative mortality and vaccination were analyzed in conjunction with surveys of seropositivity. Predictions of final mortality burden under an alternate policy scenario for Japan were calculated and the COVID-19 outcomes for China were assessed. Results: By 2025, Western countries (US, UK, Brazil and Italy) had cumulative mortality rates in the range of 3339–3548 deaths per million, about 6-fold higher than East Asian and New Zealand ‘zero-COVID’ countries. Moderate virus suppression in Japan produced the lowest cumulative mortality of the countries analyzed; if earlier policies had been maintained, the predicted cumulative mortality rate by 2025 would be one-tenth that of the US, UK, Brazil and Italy and one-half to one-third that of other zero-COVID countries. For China, transitioning from a zero-COVID policy in 2022–2023, the estimated 2025 cumulative mortality was 1607/million, half that of Western countries. Conclusions: To minimize COVID-19 mortality would require: (1) Innovation on systematic sampling of ambient airborne virus exposure to sustain low but non-zero virus levels across entire populations, and (2) seropositivity assessment (instead of mass PCR testing for new cases) for calibrating exposure management, and tracking and protecting high-risk populations.

## 1. Introduction

Across the world, national policies and practices determined the course of the catastrophic SARS-CoV-2 (COVID-19) pandemic, reflecting diverse societal cultural norms, views on basic medical issues, public health practices, resource limitations and political constraints. In the US, two popular schools of thought on pandemic management emerged: (1) positions opposed to government intervention, promoted by the then current Trump 2016 Administration (denial of pandemic, individual rights, anti-vaccination, disregard of the vulnerable), and (2) the evolving CDC/National Institutes of Health policies for which Dr. Anthony Fauci was the public presence. The primary pandemic strategy that quickly emerged in the US and Europe was rapid, large-scale vaccine development. Haphazard and inadequate primary-prevention efforts (reducing virus exposure) during early 2020, the critical pre-vaccine period, resulted in overwhelming challenges to hospital medicine and severe economic disruption. Meanwhile, a notable global pivot in basic, medical and public health research occurred (reflected in many journal offerings) producing real-time insight for pandemic policy.

A comparison of high COVID-19 mortality countries (e.g., US, U.K, Brazil, Italy) with low mortality countries (e.g., Taiwan, New Zealand, South Korea, Japan) during 2020–2024 provides an opportunity to explore the role of natural immunization in relation to the general levels of actual virus exposure over the course of pandemic mortality. This investigation would require adopting fundamental concepts in (a) environmental exposure assessment (surveys of airborne COVID-19 air concentrations) and (b) epidemiological risk characterization (infection, case identification and fatality incidence rates) in relation to airborne virus exposure. The objective here is to describe a COVID-19 prevention strategy before and after vaccine availability, based on virus air concentrations, seropositivity and mortality data, that would minimize the ultimate mortality burden and the associated social and economic disruption.

### 1.1. Early Anecdotal Evidence on Developing Immunity

In Queens NY, surveys revealed that many more people had acquired immunity than were known COVID-19 cases [1]. This important anecdotal observation was confirmed by formal investigations. Three months into the pandemic, a population-based survey of COVID-19 immunity (positive seroconversion) in Santa Clara County, CA, estimated that 53,000 people (2.8% of the population) had COVID-19 immunity with only 1200 confirmed COVID-19 cases (2.3% of infections) [2]. There were 90 attributable deaths (7.5% of cases, 0.17% of estimated total infections).

### 1.2. Early Simulation Modeling

By 1 January 2021, before vaccines, simulation modeling studies estimated that 25% of the US (about 84 million) had natural immunity from COVID-19 infection, and immunity was increasing at about 5% per month (unpublished estimates based on analyses performed by investigators at City University of New York (CUNY) [3,4] in collaboration with The New York Times [5]). The simulation modeling studies of Pei et al. [6] estimated 105 million (31% of the US population) had attained significant immunity as of 26 January 2021, almost entirely from COVID-19 infection. By 15 February 2021, the CUNY/NY Times research group estimated that 38% of the US population had acquired natural COVID-19 immunity and 4% had immunity by vaccination [3,4,5]. The 105 million estimated infections (producing immunity) corresponded to 25.3 million known cases of infection reported by CDC [7] and 400,000 deaths corresponding to a case fatality ratio (CFR) of 1.6% (400,000/25,300,000) and an infection fatality ratio (fatalities per infection, IFR) of about 0.4% (400,000/105,000,000). Only one in four infections conferring immunity were being identified, suggesting that about three-fourths of all infections were asymptomatic or low severity. Some severe COVID-19 infection cases may have been missed because of poor access to healthcare, unavailable testing or flawed reporting. By April 2022, based on surveys from all 50 states, US CDC estimated that 60% of Americans had previously experienced a COVID-19 infection (about 195 million) and less than one in three infections were being detected [8].

### 1.3. Exploring an Alternate Pandemic Strategy

Observation that most infections resulting in immunity were not reliably identified suggests (a) that low virus exposures can sufficiently stimulate the immune system to stay ahead of the developing infection (in healthy people), (b) that the exposure–response for immunity is stronger (requires less virus exposure) than that for developing serious clinical or fatal disease, and (c) that the number of new infections (conferring immunity), per fatality, may be higher in populations with lower COVID-19 exposure levels. The implications of this conjecture posed urgent research questions. For example, (1) what should be the target exposure range and the sampling regime for ambient COVID-19 air concentrations when only fleeting contact with a “heavy spreader” was likely, and (2) what target levels were appropriate for venues where sustained contact with infected individuals was likely, with exposures to larger-diameter particulate aerosols [9,10]. (A 10 µm droplet size would contain 1000 = 10^3^ times the virus copies of a 1 µm droplet.) However, research on primary prevention for COVID-19 was not a priority. The focus was on secondary prevention (early detection through extensive high-throughput testing, rapid development of COVID-19 vaccines) and tertiary prevention (improving critical care practices and capacity for COVID-19 cases).

### 1.4. Uknown Exposures and Exposure–Response Relationships

There are no studies that characterized COVID-19 inhalation exposure over time in large populations, and the appropriate exposure metrics predicting immunity or serious disease based on the time-profile of exposure are not known. However, viral load in an index case is a useful surrogate. One study bearing on the infection exposure response examined seropositivity in the 2,474,066 contacts traced from 1,064,004 presenting cases in England from 1 September 2020 to 28 February 2021 [11]. A total of 231,498 of contacts (9%) had COVID-19 PCR-seropositivity, the prevalence of which increased almost 4-fold across five levels of viral load (PCR assay) in the presenting case (*p* < 0.001). COVID-19 case mortality was also associated with viral load at diagnosis [12].

The present work explores COVID-19 seropositivity prevalence in relation to case fatality, and reviews global summary data over the course of the pandemic comparing selected countries. Predictions of final pandemic mortality burdens taking into account vaccination are calculated for comparison with (a) what was observed, (b) when final data is lacking or (c) for examining alternate scenarios. The results are discussed with respect to outstanding issues, a proposed strategy and research needs.

## 2. Methods

First, published data on pre-vaccination COVID-19 seropositivity surveys were examined in relation to estimates of relative virus exposure intensity. Second, COVID-19 cases, deaths and vaccination status were examined in selected countries with a focus on comparative cumulative mortality rates over time. Pandemic performance was assessed based on the cumulative mortality rates (per million) over the years 2020 through 2024.

### 2.1. Trends in Fatality Infection Ratio with COVID-19 Exposure Intensity

Of special interest was the *fatality infection ratio* (FIR), the number of infections conferring immunity per confirmed COVID-19 attributable death: (FIR = 1/IFR). From the IFR reported in or derivable from a number of seropositivity surveys [1,2,3,4,5,13,14,15,16,17,18,19,20,21,22,23,24,25,26,27,28,29,30,31,32], the FIR was calculated. For countries or regions in which a seropositivity survey was conducted and the associated attributable deaths were also reported, the FIR was calculated as follows (in the case of Taiwan) [13]. In July 2020, with a population of 23.6 million, the nation-wide estimate of seropositivity was 0.05% and a corresponding count of COVID-19 deaths was seven, yielding the following: FIR = infections/deaths = (0.0005 × 23,600,000)/7 = 1695 (Table 1). When national counts of corresponding COVID-19 deaths were not reported for a seropositivity survey, the cumulative COVID-19 deaths per million for that country as of the date of the survey were taken from a global database [33,34], and FIR = (infections per million)/(deaths per million) = (seropositivity × 1,000,000)/(deaths per million).

To investigate whether the FIR is higher at lower virus exposures, the relative population-average COVID-19 exposure during the survey period can be estimated making the assumption that the attributable mortality rate at a point in time is proportional to the virus exposure concentration at that time (and that most of the population remains unprotected, the rare-disease assumption). The average virus mortality rate per million in the surveyed region is the cumulative COVID-19 mortality per million in the region (up to the date of the survey) divided by the duration of observation, beginning in April 2020 when COVID-19 mortality became fulminant. This average rate was utilized as a surrogate for the relative, population-average virus exposure intensity. While this approach does not account for progress in medical mortality outcomes and is assumed linear (at the group or ecological level of analysis), it was believed to be the best measure available.

### 2.2. Extrapolation to Ultimate Pandemic Mortality Burden

Using the global COVID-19 database developed and curated at Johns Hopkins University [33] (and using a display interface [34]), cumulative pandemic mortality, IFR, FIR, CFR and vaccination status during 2020–2024 were examined in selected countries. This database does not address important demographic risk factors such as socio-economic status. Applying the COVID-19 annual mortality rate and vaccination status at the end of 2022 (Appendix A), for two more years, permitted an estimate of near-final COVID-19 mortality burden by 1 January 2025 (which was not reported for some countries or scenarios examined). In the case of Japan, abrupt increases in the COVID-19 mortality rate were observed in January and July of 2022. Hypothetical predictions of the additional deaths by 2025 were calculated based on continuation of the COVID-19 mortality rates observed prior to January or July of 2022. This prediction assumes that those COVID-19 mortality rates would be maintained with adjustments in public health practices dealing with, for example, new COVID-19 variants. Similarly, the cumulative mortality resulting by 2025 for Taiwan was estimated (due to Taiwan data no longer being reported in the global database [33,34] by 30 August 2023).

In a population, the proportion at risk of infection (unvaccinated and no prior infection, PaR), needed for predicting the final burden, can be estimated as follows:PaR = 1 − (SP × (1 − %VAC/100) × 0.90 + (1 − SP) × %VAC/100 × 0.95 + SP × %VAC/100 × 0.98)
where SP is natural COVID-19 seropositivity; SP = cumulative mortality per million [33] × FIR/1,000,000; %VAC = percent completing full original vaccination protocol [33]; fatality protection factor (PF) with prior infection is stipulated to be 90%, with vaccination is 95% and with both vaccination and prior infection is 98%. This expression is *one minus the proportion protected*, and the protected are a sum representing three distinct subpopulations: prior infected/no vaccination, no prior infection/vaccinated and prior infection/vaccinated. This calculation assumes that the FIR, even though measured in most cases prior to vaccination, applies in the future. At the end of 2024, the PaR was specified to be as follows:PaR(1 Jan 2025) = 1 − (0.95 × 0.05 × 0.90 + 0.05 × 0.95 × 0.95 + 0.95 × 0.95 × 0.98)
where the final SP and %VAC were assumed to equal 95%. The COVID-19 mortality rate in the unprotected population (TMR) was calculated from the observed overall mortality rate (OMR) at 1 Jan 2023 as follows: TMR = OMR/PaR(1 Jan 2023). In predicting the additional COVID-19 mortality during 2023–2024, the rate TMR was applied to the mean of PaR(01-01-2023) and PaR(01-01-2025):deaths (2023–2024) = population × 2 yr × TMR × (PaR(01-01-2023) + PaR(01-01-2025))/2.

## 3. Results

### 3.1. Seropositivity Surveys in 2020

Seropositivity surveys were reported from many countries during 2020 (Table 1) [1,2,3,4,5,33,34]. Some surveys were intended to be nationally representative (typically based on medical laboratory blood samples); others assessed high-risk areas. During 2020, before vaccine availability, seropositivity ranged from 0.0005 (Taiwan [13]) to 0.68 (South Africa [31] and Queens NY [1]), reflecting the non-uniform chronological spread of virus exposure, sampling strategies and diverse prevention policies and practices. Variation in the time-course of COVID-19 infections and unknown exposure levels across the surveys make interpretations with respect to FIR somewhat opaque but countries with zero-COVID policies (Taiwan, S. Korea, Japan, New Zealand) appear to have the lowest CFRs compared to UK, Germany, Sweden, Spain and Italy (Table 1). Early in the pandemic, CFRs would generally increase with improved COVID-19 death certification but decrease to a greater extent from increased testing availability over time and improved medical outcomes. The general decline in the CFR with calendar time within countries [33], by an order of magnitude in some cases (Table 2), suggests that increasing natural and vaccination immunity and lower COVID-19 exposures have played an important role in reducing the CFR in addition to the effects of more thorough case ascertainment and improvements in medical management. Interestingly, the zero-COVID countries excepting Vietnam had average rates of new COVID deaths during 2022 that were comparable to those of the Western countries studied.

### 3.2. Seropositivity and COVID-19 Exposure Intensity

In the countries and regions with concurrent seropositivity surveys and cumulative fatality data, the evidence supports the conjecture that the FIR decreases with increasing average COVID-19 mortality rate (surrogate for COVID-19 exposure). In the Western countries with high COVID-19 mortality, the mean relative exposure measure was 119 with corresponding geometric mean FIR of 179 (Appendix A). The East Asian and New Zealand countries had a mean relative exposure measure of 0.47 with corresponding geometric mean FIR of 1466, but some seropositivity measures were very early in the pandemic and probably unrepresentative. Plots of FIR with relative exposure intensity reveal a trend of decreasing FIR by an order of magnitude or more over the range of increasing relative exposure (Appendix A), but the data are limited.

### 3.3. Comparative Trends in National Cumulative COVID-19 Mortality

A comparison of the actual time course of COVID-19 cumulative mortality rates across countries provides further insight with implications for optimum pandemic management [33,34]. Most large or developed Western countries performed poorly on COVID-19 mortality, e.g., US, UK, Brazil, Italy and Russia (Figure 1). By the end of 2024, the US, UK, Italy and Brazil had sustained remarkably similar cumulative COVID mortality ranging 3339–3548 deaths per million (Table 3). In contrast, New Zealand, Vietnam, Japan, Taiwan and South Korea, which had quickly implemented stringent (zero- COVID) exposure controls, exhibited very low COVID-19 mortality (Figure 1 and Figure 2). Their cumulative death rates ranged 433–694 per million by the end of 2024, 6-fold lower than in the higher-risk Western countries (Table 3). China executed an extreme lockdown policy extending through 2022, which was abandoned or heavily revised in Dec 2022. Data availability and reporting criteria may have changed in China over time and COVID-19 mortality prior to 2023 are not analyzed in detail here.

### 3.4. Patterns of Mortality Rates in Relation to Apparent Changes in COVID-19 Exposure

Comparing the time course of COVID-19 deaths in the East Asian and New Zealand examples suggests important insights. Taiwan experienced the lowest mortality rate of any country until about July 2021. After seeing repeated minor spikes where 99.7% of new cases are asymptomatic or mild, Taiwan (apparently briefly) relaxed controls to achieve higher COVID-19 natural immunity (Figure 2) [35]. As a result, cumulative COVID-19 mortality advanced to a new plateau still far below almost all other countries. Then, in May 2022, it appears that a further relaxation and/or advent of new variants preceded a sustained large increase in the COVID-19 annual mortality rate (slope of cumulative deaths curve, Figure 1 and Figure 2) which, at the end of 2022, exceeded the COVID annual mortality rates in the US and UK (but less than rates in Italy and Germany) (Table 4). At the end of 2022, Taiwan’s COVID-19 cumulative mortality (deaths per million) was the highest of the zero-COVID countries analyzed here (Table 4, Figure 2). This history reveals how effective zero-COVID policies were in preventing deaths but also in avoiding natural immunity, and leaving citizens without immunity at high risk from endemic COVID-19 exposures. By the end of 2024 (for which current mortality data for Taiwan was unavailable), based on the annual COVID mortality rate at the end of 2022, the total domestic COVID-19 deaths were predicted to be 27,895 (1182 × 23.6, Table 4), which (at the same cumulative mortality rate) would be equivalent to 388,878 deaths in the US (vs. 1,167,292 reported).

New Zealand, with a zero-COVID policy and lockdown like Vietnam and China, exhibits a similar history with a single upward escalation in mortality rate occurring abruptly in Mar 2022 (Figure 2). South Korea experienced modestly higher mortality beginning earlier (Jan 2021) but much higher starting in Feb 2022 and by the end of 2022 a cumulative mortality rate very close to that of Taiwan, and a mortality rate (slope of cumulative rates) comparable to Taiwan and New Zealand (Figure 2). The elevated mortality rates toward the end of 2022 in all the zero-COVID countries (Table 4) appear to reflect the low prevalence of immunity (natural or vaccine) and account for the high overall annual COVID mortality rate observed for those countries for the year 2022 (Table 2).

Of the better-performing countries analyzed, Japan experienced higher mortality rates sooner, beginning in November 2020, but by November 2022 had the lowest cumulative COVID-19 mortality of any of the countries analyzed here. By the end of 2021, Japan and Vietnam had permitted the highest cumulative mortality rate (150 and 300/million, respectively) but had the lowest cumulative mortality at the end of 2022 (Figure 2). At the end of 2022, New Zealand had the same low cumulative rate but it was increasing rapidly as in S. Korea and Taiwan. These observations imply that higher early COVID-19 exposures (and mortality) in Japan were promoting higher immunization rates than in other zero-COVID countries but achieving lower cumulative mortality rates by the end of 2022, much lower than in the West. However, Japan experienced two abrupt increases in mortality (in January and July 2022). Without them, the cumulative mortality would have been substantially lower by end of 2022 (Figure 2). Vietnam appears to have relaxed precautions in about July 2021 with a corresponding leap in the mortality rate until April 2022 when either strong restrictions were re-imposed, reporting of COVID-19 mortality was curtailed or vaccination was largely complete. Vietnam had the highest vaccination prevalence of all these countries (92% in December 2022: 87% full initial protocol, 5% partial) and reported the lowest cumulative mortality by end of 2022 (Figure 2).

### 3.5. Effect on Acquired Natural Immunity of Prolonging Zero-COVID Controls

The East Asian/New Zealand countries with the highest final cumulative deaths per million were those that appeared to maintain zero-COVID policies the longest, relaxing controls later in time (Taiwan, South Korea, New Zealand) compared to Japan (Figure 2). A reasonable inference from the zero-COVID country comparisons is that there is a range of COVID (airborne) exposure and associated low but significant mortality that ultimately results in the lowest cumulative mortality burden.

At the end of 2022, among all the countries analyzed, Japan had the highest COVID-19 *annual* mortality rate, 986/million (Table 4). However, subsequent policy adjustments lowering COVID-19 mortality rates appear to have been implemented (Appendix A). The predictions here of final burdens (for all the countries for which the final cumulative mortality was also reported [33]) were greater than was actually observed (Appendix A) probably because the annual rate applied in the estimation algorithm for the period 2023–2024 was too high (based on the rates observed at the end of 2022). For the two counterfactual scenarios in Japan maintaining lower mortality rates (beginning in January and July 2022 and observed over periods of six months), the prediction algorithm yielded final cumulative mortality estimates of 348/million and 276/million, respectively, much lower than the observed final rate, 598/million. The hypothetical Japanese cumulative mortality outcomes were exceptionally superior to that of the Western countries (by a factor of 10) and were one-half to one-third of the mortality burdens of other zero-COVID countries (Table 4). These observations support the conclusion that an optimum COVID-19 control strategy would maintain a low but non-zero exposure level both before and after the advent of vaccination. The levels achieved by Japan (and possibly Vietnam) over the period 2020–2021 appear to have been in the optimum range.

### 3.6. The Chinese COVID-19 Pandemic Experience

Analyses on the policy changes implemented in China in late 2022 are revealing. Based on reported provincial cremation data, estimates of COVID-19-related mortality over time were calculated [36]. Making assumptions that infections rose relatively uniformly across many provinces, the national toll during the 2022–2023 surge was estimated to be 1.5 million [36]. Multiple investigators (Ben Cowling, Univ. Hong Kong; Lauren Ancil Meyers, Univ. of Texas, Austin, and Zhanwei Du, Univ. Hong Kong; Yong Cai, Univ. North Carolina, Chapel Hill) also estimated that the surge in mortality beginning in late 2022 comprised about 1.5 million deaths and that about 80–90% of the Chinese population had been infected [36]. Investigators Du and Wang et al., using Chinese infection and vaccination data, estimated 1.2–1.7 million COVID-19 deaths during the surge [37]. The estimates for the surge deaths from independent sources add to the credibility of the 1.5 million cremation-based estimate. An interpretation of the Chinese experience based on the present work suggests that large numbers of the Chinese population lacked COVID-19 immunity until late 2022 and thus were highly vulnerable following the policy change, especially those in high-risk populations. It is unclear to what extent, if any, protective behaviors and policies continued during the surge.

Applying to China the cumulative mortality rates of the other zero-COVID countries analyzed here (about 540/million by the end of 2022) for a population of 1.4 B (Table 5), implies that about 750,000 total Chinese COVID-19 deaths (540/million × 1400 M) occurred prior to the surge. Therefore, the cumulative COVID-19 mortality by the end of the surge was probably about 2.25 million (1.5 M + 750 K), implying a final burden (cumulative mortality per million) of 1607 (2,250,000/1400 M), 0.16% of the population (compared to 0.34% in the US). Applying the US COVID-19 cumulative mortality rate to China implies 4.8 M COVID-19 deaths would have occurred (2.25 × 0.34/0.16), suggesting the harsh Chinese zero-COVID policies may have avoided about 2.5 M deaths. The Chinese experience in a large population with rigorous virus suppression supports the inferences drawn here from the Western and East Asian/New Zealand countries analyzed. A more nuanced transition at the end of 2022 and a public health strategy from the outset optimizing natural and vaccination immunity perhaps could have perhaps avoided another one million deaths in China with much less economic trauma.

### 3.7. Drivers of Pandemic Control Decisions

The COVID-19 *annual* mortality rates at the start of 2023 were higher in Italy, Germany and Taiwan, but highest (perhaps briefly) in Japan: 986/million (Figure 1 and Figure 2 (slope); Table 4) which was more than double those of New Zealand, South Korea and Taiwan. The observations suggest that relaxation of controls likely resulted in dramatic increases in mortality rates among the unprotected (Table 5: Annual mortality rate/million... population-at-risk). The basis for policy relaxations in the better-performing countries may have included considerations of real-time overall COVID-19 mortality rates. These rates were calculated on *full* populations, not those at risk (unvaccinated with no prior infection). If immunity from infection and vaccination was not accounted for, relaxing controls would have placed sharply escalating risk on those unprotected. The final (unprotected) rate at the end of 2022 in Japan (5245/million, Table 4) approaches that of the UK during its early crisis period (6751/million) (Table 5).

In China, with an estimated 80–90 percent immunity after the surge [36], if there was 20% immunity before the surge, then during the surge about 65% (85–20) of the population acquired immunity during 3 months. There were 1.5 million COVID-19 deaths during the surge [36,37], implying about 600 infections per death (FIR = 1.4 B × 0.65/1.5 M = 607), and indicating that most immunizations probably occurred at low virus exposures in a country coming out of extreme COVID-19 suppression. The high surge mortality rate among the unprotected in China (>5000/million per year, Table 5) may reflect a very non-uniform distribution of COVID-19 exposure, e.g., urban vs. rural areas, with urban areas having higher mortality and an FIR much lower than 607.

## 4. Discussion

### 4.1. Conclusions

Although Japan appears to have achieved a high level of success, this result may have been fortuitous. No programmatic incorporation of ambient airborne exposure information was evident in any country for identifying target exposure levels. Exposure assessments were conducted only in limited locations such as hospitals [38]. The superior performance of the East Asian countries studied and New Zealand may reflect important public health policy choices but also other critical factors such as the widespread acceptance and routine use of mask PPE and public trust in mandated preventative practices.

#### 4.1.1. Specific Key Observations

Taiwan and possibly some other countries achieved very low COVID-19 mortality rates without resorting to comprehensive lockdowns.The countries or regions maintaining very low COVID-19 mortality rates also largely suppressed natural COVID-19 immunity.Widespread acceptance of face masks in the zero-COVID countries suggests this is a critical component of exposure control along with distancing and isolation procedures.The policies and practices followed in Japan during 2020–2021 appear to have been close to optimal for minimizing the COVID-19 pandemic mortality burden.Decisions to relax strong COVID-19 suppression based on diminishing case or fatality rates resulted in high mortality rates among residual unprotected populations.

#### 4.1.2. The Findings Here Support These Recommendations

A primary prevention objective should be to systematically monitor ambient COVID-19 air concentrations to manage low but non-zero levels in an optimal range. Supplemental sampling protocols would target: educational, retail, transportation, medical, recreational and workplace [39] environments. Sampling the built environment would inform distancing, filtration and air-change goals. Efficient, low-cost sampling and high-throughput determination of airborne virus concentrations would be promoted. Downsizing routine respiratory protection based on exposure surveys from full cannister to N95 facemasks and primitive cloth masks would be appropriate. Most environmental toxigens are regulated by way of a maximum allowed concentration, even carcinogens believed to have “no safe level.” Viral infections like COVID-19 and other infectious epidemic diseases are anomalous in that zero-exposure is considered an ideal objective even though lacking a scientific basis in pathophysiology, risk assessment or public health strategy.

Assessment of seropositivity prevalence [40,41] should place routine, repetitive mass (costly) PCR testing for new cases. Seropositivity information would be needed to (a) specify the operational parameters of the exposure control strategy, (b) inform secondary prevention (e.g., allocation of masks), (c) identify the remaining unprotected population and d) facilitate decision-making in the workplace [42,43,44,45,46,47,48,49]. Seropositivity surveys would be conducted on small representative samples of local populations, but for some subpopulations would attempt to survey everyone: nursing homes, anyone having routine contact with the public or co-workers. Travelers by air or rail could be required to show seropositivity documentation (or a current negative PCR test). High throughput seropositivity testing would drive low-cost innovations as happened with PCR, lateral flow and other testing for COVID-19 cases.

### 4.2. Controversy Early in COVID-19 Pandemic

Within 6 months of the global spread of COVID-19 infections, two soundly contrary proclamations were issued in the public health realm [50,51]. The contention was miss-characterized as being over the pandemic strategy “herd immunity” but in reality, herd immunity is not a strategy, it is a condition. All arguable strategies achieve herd immunity by means of some combination of natural immunity and vaccination. The Great Barrington [GB] Declaration [50] argued against draconian restrictions and blanket lockdowns claiming: “Keeping these measures in place until a vaccine is available will cause irreparable damage, with the underprivileged disproportionately harmed. As immunity builds in the population, the risk of infection to all—including the vulnerable—falls... We know that all populations will eventually reach herd immunity—i.e., the point at which the rate of new infections is stable—and that this can be assisted by (but is not dependent upon) a vaccine. Our goal should therefore be to minimize mortality and social harm until we reach herd immunity.” The Declaration called for strong primary prevention policies to protect vulnerable and general populations, policies that exceeded the intent of some of those being routinely advocated or applied. The John Snow [JS] Memorandum [51] dismissed a significant role for natural immunity (“Any pandemic management strategy relying upon immunity from natural infections for COVID-19 is flawed.”) even though that was the *only* route to immunity during the critical year 2020 and into 2021. The JS statement suggested without evidence that natural immunity would quickly dissipate. It falsely implied absence of primary prevention in the GB proposal (“Uncontrolled transmission in younger people risks significant morbidity and mortality across the whole population.”).

Although the GB Declaration assigns a central role for natural immunity in early pandemic management, neither it nor the JS Memorandum (nor U.S. CDC or other public health authorities) conceived of an evidence-based primary prevention strategy utilizing airborne exposure assessment and seropositivity surveys. However, the GB Declaration did propose seropositivity testing for managing risk in workers with routine public or co-worker contact.

### 4.3. Equity Considerations and Generalization in Pandemic Management

Environmental justice should be examined in the pandemic context as communities at high risk due to housing, air pollution, transportation and other demographic factors are also sources of general contagion. Many workers reside in high-risk communities including those deemed “essential” but afforded minimal protection [52,53,54,55].

The early manifestations of avian virus (H5N1) human contagion in 2024 present an opportunity for epidemic management based on environmental science [56,57]. Rather than widespread testing of entire dairy herds by way of combined milk-sample determinations, efficient sampling strategies for airborne virus would permit early identification of herds with an infection. Prompt interventions would follow to protect farm workers, identify and isolate infected cows, and promote natural immunization of herds while avoiding high exposures. Similarly, there are implications for H5N1 management in other animal populations (e.g., chickens) for promoting natural immunity and avoiding mass culling of farm populations. Initial animal seropositivity studies associated with H5N1 air concentrations would be needed.

### 4.4. Strengths, Limitations and Needs

The countries with zero-COVID policies focused on here (in East Asia and New Zealand) all have modern public health infrastructure comparable or superior to that of the high-risk countries (e.g., UK, US, Italy, Brazil). The pandemic database curated at Johns Hopkins University is an exceptional resource but the present review has multiple unavoidable limitations: (a) paucity of published seroprevalence information with corresponding cumulative fatality data, (b) incomplete or unrepresentative COVID-19 case and fatality identification over time, (c) total absence of virus exposure information and (d) assumption of a quasi-linear exposure–response relationship between virus exposure and attributable mortality incidence. This study’s reliance on ecological-level data is a fundamental limitation, especially the implied uniformity of exposure within target populations, even entire countries. The study data-reporting limitations imply significant quantitative bias early in the pandemic but likely would have had a small impact on final cumulative mortality rates/million with increasing alignment of reporting protocols as the pandemic progressed. These biases could not have accounted for most of the wide disparities observed between Western and East Asian societies.

Statistical variability generally was not an issue because, for the measures of seropositivity, cases and fatalities, the observed numbers were large except in the early days of the pandemic in zero-COVID countries. The choice of countries to study was restricting but included important examples contrasting mortality outcomes; no alternate country choices were investigated and discarded. For countries with limited public health resources, the available data would not have been adequate for this analysis. The conclusions presented need validation and elaboration within the public health and broader research community, including pandemic modeling.

## Figures and Tables

**Figure 1 ijerph-22-01145-f001:**
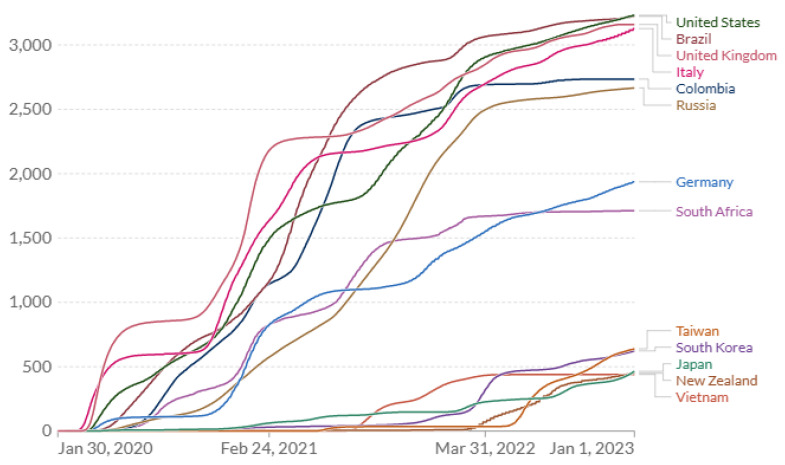
Cumulative confirmed COVID-19 deaths per million for selected countries (data from Johns Hopkins University CSSE [33], displayed by software at OurWorldInData.org [34] and provided under Creative Commons BY open access license).

**Figure 2 ijerph-22-01145-f002:**
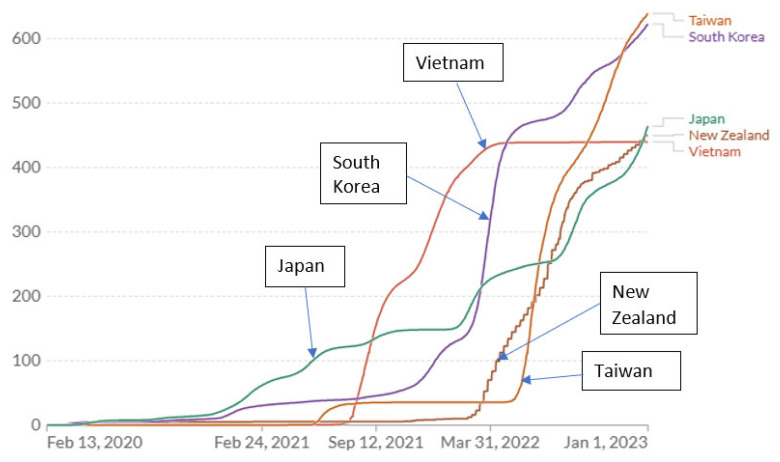
Cumulative confirmed COVID-19 deaths per million for East Asian countries and New Zealand, which exhibited low death rates (data from Johns Hopkins University CSSE [33], displayed by software at OurWorldInData.org [34] and provided under Creative Commons BY open access license).

**Table 1 ijerph-22-01145-t001:** COVID-19 seropositivity surveys prior to vaccination availability in order of increasing seropositivity.

Country/Region [1,2,3,4,5,13,14,15,16,17,18,19,20,21,22,23,24,25,26,27,28,29,30,31,32]	Survey Date mm/dd/yyyy	Seropositivity	IFR ^1^	FIR ^1^	CFR ^1^
Taiwan	07-15-2020	0.0005	0.00059	1695 ^2^	0.0153
New Zealand	12-16-2020	0.0010	0.0050	200	0.0124
S. Korea	11-01-2020	0.0024	0.0037	268	0.0175
Vietnam/high risk region	10-15-2020	0.004	0.00009 ^2^	11,111 ^2^	-
Germany/7 regions	08-15-2020	0.020	0.015	67	0.040
US/California (Santa Clara)	03-20-2020	0.028	0.0017	588 ^2^	0.075
Japan/Kobe	04-07-2020	0.033	0.00002 ^2^	42,857 ^2^	0.021
Spain	07-06-2020	0.052	0.0115	87	0.113
Sweden	07-23-2020	0.073	0.00736	136	0.075
Germany/Tirschenreuth	11-22-2020	0.092	0.023	43	0.084
Brazil/Matinhos	07-01-2021	0.11	-	-	-
UK/Southeast	05-12-2020	0.12	-	-	-
Iran/Guilan	04-15-2020	0.12	-	-	-
UK	01-15-2021	0.15	0.0108	93	0.034
India/Ahmedabad	12-31-2020	0.18	-	-	-
US/New York State	06-15-2020	0.22	-	-	-
Italy	05-10-2020	0.23	0.0022	455	0.140
US/New York City	06-26-2020	0.26	-	-	-
Saudi Arabia/Jazan	11-01-2020	0.26	-	-	-
Yemen/Aden	12-01-2020	0.27	-	-	-
US	12-31-2020	0.31	0.0033	303	0.0173
Colombia/Monteria	10-14-2020	0.55	0.0058	172	0.063
US/NYC (Queens)	06-15-2020	0.68	0.0021 ^2^	476 ^2^	-
SA/Gauteng	12-15-2020	0.68	0.00061 ^2^	1639 ^2^	0.028

IFR—infection fatality ratio; FIR—fatality infection ratio (1/IFR); CFR—case fatality ratio; ^1^ When prior COVID-19 cases and fatalities are known on date of seropositivity survey; when day of month not reported, 15 assigned; ^2^ Likely under-ascertainment of COVID-19 deaths.

**Table 2 ijerph-22-01145-t002:** Annual average COVID-19 case fatality ratio (CFR) by country and year [33], in order of 2020 deaths per million.

Country	Year	Cases per Million	Deaths per Million	CFR
UK	2020	36,865	1407	0.0382
	2021	154,782	1223	0.0079
	2022	165,862	529	0.0032
US	2020	59,763	1036	0.0173
	2021	102,538	1406	0.0137
	2022	135,553	789	0.0058
Brazil	2020	35,673	906	0.0254
	2021	67,859	1971	0.0290
	2022	65,205	346	0.0053
Japan	2020	1902	28.2	0.0148
	2021	12,082	119	0.0099
	2022	221,871	316	0.0014
S. Korea	2020	1192	17.7	0.0148
	2021	11,068	92.3	0.0083
	2022	549,669	513	0.0009
New Zealand	2020	417	5.0	0.0120
	2021	2306	4.3	0.0019
	2022	401,277	440	0.0011
Taiwan	2020	33.4	0.29	0.0087
	2021	679.6	35.3	0.0520
	2022	369,572	603	0.0016
Vietnam	2020	14.9	0.36	0.0242
	2021	17,617	331	0.0188
	2022	99,749	108	0.0011

**Table 3 ijerph-22-01145-t003:** Cumulative confirmed COVID-19 cases per thousand and deaths per million [33] on 1 July 2020 and 31 December 2024, in order of decreasing cumulative COVID-19 deaths per million on 1 July 2020.

Country	Cum. Cases/1000	Cum. Deaths/Million	CFR ^1^	Cum. Cases/1000	Cum. Deaths/Million	CFR
	as of 1 July 2020			as of 31 December 2024		
UK	4.20	834	0.199	367	3404	0.009
Italy	4.08	589	0.144	452	3345	0.007
US	8.14	381	0.047	303	3548	0.012
Brazil	6.98	289	0.041	178	3339	0.019
Germany	2.34	108	0.046	457	2081	0.005
S. Africa	2.81	47.5	0.017	65.2	1645	0.025
Japan	0.153	7.88	0.052	270	598	0.002
S. Korea	0.250	5.44	0.022	668	694	0.001
New Zealand	0.295	4.44	0.015	519	876	0.002
Taiwan ^2^	0.018	0.29	0.016	-	-	-
Vietnam ^3^	0.003	-	-	117	433	0.004

^1^ Case fatality ratio (CFR) based not on individual case outcomes but reporting of total numbers; deaths in general would have occurred at some time after certification of cases. ^2^ Taiwan excluded from OurWorldInData.org [34] at some time after 2022. ^3^ Very small numbers of deaths reported for Vietnam in 2020.

**Table 4 ijerph-22-01145-t004:** COVID-19 mortality rates at end of 2022 and final pandemic mortality burden by end of 2024 as well as predicted burden in (a) Japan under two alternate scenarios and (b) for Taiwan (missing data).

Country	Pop (M)	Cum. Mort. Rate/Million @ End 2022	Annual Mort. Rate/Million @ End 2022 ^1^	% Vacc. @ End 2022	Avg. FIR	Proportion at Risk @ End 2022 (Prevalence)	Annual Mort. Rate/Million @ End 2022 in Population at-Risk ^2^	Cum. Mort. Rate/Million @ End 2024	Pandemic Deaths @ End 2024
US	329	3230	367	68.6	227 ^3^	0.126	2913	3548	1,167,292
UK	67.1	3159	367	76.5 ^4^	94	0.204	1799	3404	228,408
Italy	59.5	3128	573	81.3	94 ^5^	0.171	3351	3345	199,028
Germany	83.2	1937	520	76.5	75	0.239	2176	2081	173,139
Taiwan ^7^	23.6	639	460	86.3	266 ^6^	0.155	2968	1182 ^7^	27,893
S. Korea	51.7	623	393	86.3	266	0.155	2535	694	35,880
New Zealand	5.1	450	267	79.8	200	0.223	1197	876	4468
Japan	126.3	464	986	83.2	266 ^6^	0.188	5245	598	75,527
Japan ^8^	126.3	235	80	82.0	266	0.209	383	348 ^8^	43,952
Japan ^9^	126.3	145	80	81.0	266	0.223	359	276 ^9^	34,859

^1^ Annual rate based on 9-month period (31 March 2022 to 31 December 2022) from graphical presentation (Figure 1 and Figure 2); during 2022–2024 mortality rate at end of 2022 applied to at-risk subpopulation (unvaccinated or zero seropositivity) assuming that group declines to almost zero by end of 2024, with protection factors: PF(seropos.) = 0.80, PF(vacc.) = 0.90 and PF = 0.98 for both natural and vaccine immune protection; final annual mortality rate not readily derivable from OurWorldInData.org [34] at end of 2024. ^2^ For US: 2913 = 367/0.126. ^3^ US FIR based on IFR = 0.00441 derived from Clarke et al. [8]. ^4^ Vaccination status for England. ^5^ FIR for Italy (444, 10 May 2020) replaced by value from UK: 94. ^6^ FIR for Japan (42,857, very early in pandemic, 7 April 2020) and for Taiwan (1695, 15 July 2020) replaced by value from S. Korea: 266, 1 November 2020). ^7^ Data absent from OurWorldInData.org [34] in 2024; predicted based on annual mortality rate observed at end of 2022. ^8^ Prediction based on cumulative deaths and mortality rate as of 24 July 2022. ^9^ Prediction based on cumulative deaths and mortality rate as of 27 January 2022.

**Table 5 ijerph-22-01145-t005:** Initial and 2022 COVID-19 mortality rates per million population.

Country	Pop (M)	Initial Annual Mortality Rate/Million ^1^	Annual Mortality Rate/Million ^2^ @ End 2022	Cum. Mort. Rate/Million @ End 2022	Proportion at Risk ^3,4^ @ End 2022 (Prevalence)	Annual Mortality Rate/Million in Population at Risk ^4^ @ End 2022
US	329	2591	367	3230	0.126	2913
UK	67.1	6751	367	3159	0.204	1799
Taiwan	23.6	1.19	460	639	0.155	2968
S. Korea	51.7	14.3	393	623	0.155	2535
Japan	126	<10	986	464	0.188	5245
estimate from July 2022		<10	80 ^4^	235	0.209	383
estimate from February 2022		<10	80 ^4^	145	0.223	359
China, 0% immunity ^3^	1400	<181 ^5^	<181 ^5^	<544 ^5^	1.0	5477 ^6^
China, 10% immunity ^3^		<181 ^5^	<181 ^5^	<544 ^5^	0.9	5893 ^6^
China, 20% immunity ^3^		<181 ^5^	<181 ^5^	<544 ^5^	0.8	6368 ^6^

^1^ During first 4 months of 2020 (Taiwan, during first 3 months). ^2^ As of end of 2022. ^3^ As of end of 2022: final annual mortality rate divided by population at risk, e.g., for US: 367/0.126 = 2913. For China, hypothetical population at risk (prevalence) prior to surge on 1 January 2023, which is assumed to decline exponentially during surge of 2023 to 0.1 (90% protected) by 1 April 2023. ^4^ As of end of 2022, but for Japan without two relaxation implementations, as of 24 July 2022 and 27 January 2022; for China during surge of early 2023. ^5^ China cumulative COVID-19 mortality per million in 3-year period 2020–2022 estimated based on cumulative rates from the zero-COVID countries analyzed as of end of 2022 (Table 4, using mean of 639,623,450,464 = 544); an over-estimate because those countries experienced surges beginning in 2022. Estimated initial and final (average) Chinese annual rates: 3-yr cum. rate/3 = 544/3 = 181. ^6^ In China, average COVID-19 annual COVID-19 mortality rate during 3-month surge, Jan–Mar 2023: 1,500,000 × 4/1400 = 4286/million; final fixed rate during surge.

## Data Availability

All of the data in this study are in the public domain and the sources are cited.

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
