# Peer review of "The Global COVID-19 Pandemic Experience: Innovation Through Environmental Assessment and Seropositivity Surveillance"

_ijerph, 2025, doi:10.3390/ijerph22071145_

Round 1

Reviewer 1 Report

Comments and Suggestions for Authors

Interesting study, I do have a few observations/comments

1) Were the different strains of COVID-19 documented in the study? Were they recorded in the data you analyzed?

2) I would be interested in the amount of health expenditures that each of the analyzed nations spent on preventing and treating COVID-19. Was that information available? I know in New Zealand they implemented more stringent measures, which may have lead to initial reduction in cases but lead to a higher than expected economic downturn

3) Did the vaccine statistics only recorded who took the initial vaccination...were booster rates also recorded?

4) On COVID-19 related deaths, were comorbidities also indicated?

Author Response

Reviewer comment

Author’s response

Revised text

1) Were the different strains of COVID-19 documented in the study? Were they recorded in the data you analyzed?

No, they were not. As of Jun 18, 2025, the Johns Hopkins/OWD database lists no displays that address COVID-19 variants or strains, and no such data was available for this study.

2) I would be interested in the amount of health expenditures that each of the analyzed nations spent on preventing and treating COVID-19. Was that information available? I know in New Zealand they implemented more stringent measures, which may have lead to initial reduction in cases but lead to a higher than expected economic downturn

There are many important questions regarding how national or local policies affected   the pandemic mortality outcomes that were analyzed here over time, but these questions were far beyond the scope of this investigation. The distinct time course of the New Zealand case suggested important insights that are discussed in this paper. There is some broad classification of national policies in the OWD data such as restrictions on gatherings or on transportation but comparability of methodologies across areas and over time would be difficult to assess. I don’t see any data on national expenditures in the OWD database.

3) Did the vaccine statistics only recorded who took the initial vaccination...were booster rates also recorded?

The OWD database presented vaccination status prevalence over time for the initial protocol contact and for completion of the protocol but it is not specific regarding vaccination regimens or various booster stages.

4) On COVID-19 related deaths, were comorbidities also indicated?

The OWD data does not include comorbidities but does have data on total excess mortality (observed mortality vs. what would be expected in the absence of COVID-19) which would be another important outcome for estimating pandemic burden. Excess mortality would potentially be very useful for early, real-time characterization of the pandemic natural history prior to vaccination. This outcome is independent of any COVID-19 reporting standards. That would help inform prevention policies including design of virus air sampling surveys and establishing target exposure objectives as recommended here. I have not worked with this excess mortality data from the OWD database.

Reviewer 2 Report

Comments and Suggestions for Authors

This manuscript addresses a timely and underexplored issue with creative thinking and potentially valuable insights. However, significant revisions are necessary to enhance its scientific rigor and communicative clarity. I recommend revision with a focus on improving methodological transparency, linguistic clarity, and the integration of current literature.

Many sections would benefit from clearer topic sentences and transitions. For example, in lines 54–56 (“The objective here is to define a strategy…”), clarify early on that this paper seeks to evaluate policy outcomes across countries using mortality and seropositivity data.

The paragraph beginning on line 57 shifts rapidly from anecdotal observations to modeling estimates. It might be clearer to split this into two: one discussing initial seroprevalence findings, and another on modeled estimates.

The method used to calculate “average COVID-19 exposure intensity” (lines 126–130) is not well justified. Please explain why cumulative mortality divided by time and population is considered a valid proxy, and acknowledge its limitations (e.g., it does not account for differences in testing rates, population age structures, or healthcare systems).

In Table 1, the FIR values for countries are derived but the source methods are not described. For instance, the FIR for Taiwan is listed as 17,242, based on a seroprevalence of 0.0005. The equation for this and assumptions (e.g., completeness of death reporting) should be explicitly included in the Methods section.

Several sentences are excessively long and would benefit from being split or simplified. Example:

(lines 82–85): “Observation that most infections resulting in immunity were not reliably identified suggests that low virus exposures can sufficiently stimulate the immune system…”

The phrase “it will be argued here…” (line 48) is better suited to an opinion essay than a scientific article. Consider rephrasing as: “This analysis explores whether…”

The assertion that Japan’s policy was “close to ideal” (line 285) should be framed more cautiously. This conclusion is based on retrospective modeling and assumptions about exposure levels that are not directly measured.

In Section 3.3, the Chinese mortality estimate relies on cremation data (line 288), but no reference to how this data was validated is provided. Including citations for these methods and discussing potential biases would improve credibility.

The manuscript could engage more deeply with recent literature on serological surveillance. For example, how does the proposed approach compare with established models for targeted serosurveys (e.g., in workplace settings or sentinel sites)?

The discussion on mask efficacy and ambient virus exposure (lines 368–372) could be supported with citations from aerosol transmission studies or guidelines by WHO/CDC.

Line 36: “denial of pandemic” → revise to “pandemic denial” or “denial of the pandemic.”

Line 79: Consider rephrasing “CDC estimated that 60% had been infected” to clarify that this refers to estimated cumulative incidence, not active infection.

Table 2: Label units consistently (e.g., “Deaths per million” instead of just “Deaths/M”)

Comments on the Quality of English Language

The manuscript is generally written in comprehensible academic English, but the quality of language can be improved for clarity, conciseness, and fluency:

  • Sentence Complexity: Many sentences are too long and complex, making the text difficult to follow.

  • Redundancy: Repetitive phrasing and unnecessary words reduce clarity.

  • Ambiguous References: Vague use of pronouns like "this" or "it" can confuse readers.

  • Tone and Formality: Some informal or conversational phrases need to be revised to meet academic standards.

  • Minor Grammar Issues: Occasional errors in article usage and subject-verb agreement are present.

Author Response

Reviewer 2

This manuscript addresses a timely and underexplored issue with creative thinking and potentially valuable insights. However, significant revisions are necessary to enhance its scientific rigor and communicative clarity. I recommend revision with a focus on improving methodological transparency, linguistic clarity, and the integration of current literature.

Many sections would benefit from clearer topic sentences and transitions. For example, in lines 54–56 (“The objective here is to define a strategy…”), clarify early on that this paper seeks to evaluate policy outcomes across countries using mortality and seropositivity data.

Agreed.

The objective here is to describe a COVID-19 prevention strategy before and after vaccine availability, based on virus air-concentrations, seropositivity and mortality data. e intent would be to minimize the ultimate mortality burden and the associated social and economic disruption.

The paragraph beginning on line 57 shifts rapidly from anecdotal observations to modeling estimates. It might be clearer to split this into two: one discussing initial seroprevalence findings, and another on modeled estimates.

Agreed.

New heading: 1.1 Early anecdotal evidence on developing immunity

New heading and revised text: 1.2 Early simulation modeling

By Jan 1 2021, before vaccines, simulation modeling studies estimated 25% of the U.S. (about 84 million) to have natural immunity from COVID-19 infection, and immunity was increasing at about 5% per month (unpublished estimates based on analyses performed by investigators at City University of New York (CUNY) 3,4 in collaboration with The New York Times5). The modeling studies of Pei...

The method used to calculate “average COVID-19 exposure intensity” (lines 126–130) is not well justified. Please explain why cumulative mortality divided by time and population is considered a valid proxy, and acknowledge its limitations (e.g., it does not account for differences in testing rates, population age structures, or healthcare systems).

This difficult methodological segment has been revised to improve clarity and in the first submission actually incorrectly described the method that was used to estimate the exposure surrogate. This has been corrected. Thank you for prompting this revision.

Although the exposure estimate depends on assuming a linear relationship with mortality, actually, what is of interest is not linearity, just a monotonically increasing FIR with lower virus exposure.

Revised text: To investigate whether the FIR is higher at lower virus exposures, data on seropositivity (positive COVID-19 seroconversion) and COVID-19 mortality were examined.  The infections occurring in the survey period of COVID exposure prior to a seropositivity assessment were derived as: seropositivity prevalence × population surveyed.  The relative population-average COVID-19 exposure during that period can be estimated making the assumption that the attributable mortality rate at a point in time is proportional to the virus exposure concentration then (and that most of the population remains unprotected, the rare disease assumption). The average mortality rate in the surveyed region can be calculated as the cumulative COVID-19 mortality per million in the region (up to the date of the survey) divided by the duration of observation beginning in April 2020 when COVID-19 mortality became fulminant). This average rate was utilized as a surrogate for the relative, population-average virus exposure. While this approach does not account for progress in medical mortality outcomes and is assumed linear (at the group or ecological level of analysis), it was believed to be the best measure available.

In Table 1, the FIR values for countries are derived but the source methods are not described. For instance, the FIR for Taiwan is listed as 17,242, based on a seroprevalence of 0.0005. The equation for this and assumptions (e.g., completeness of death reporting) should be explicitly included in the Methods section.

Agreed. The IFR calculated for Taiwan declined slightly when I used an additional significant figure in the cumulative mortality rate: going from 0.29 to 0.299.

Revised text: An IFR was reported in or derivable from a number of seropositivity surveys. 1-5,15-34 For countries or regions in which a seropositivity survey was conducted and the associated attributable deaths were also reported, the IFR was calculated as follows (in the case of Taiwan).15 In July 2020 with population 23.6 million, the nation-wide estimate of seropositivity was 0.05% and a corresponding count of COVID-19 deaths was seven, yielding: IFR = deaths / infections  =  7 /(0.0005 × 23600000 ) = 0.00059 and FIR = 1/IFR = 1695 (Table 1). When national counts of corresponding COVID-19 deaths were not reported for a seropositivity survey, the cumulative COVID-19 deaths per million for that country as of the date the survey were taken from the global database,13,14 and IFR = (deaths per million) / (infections per million) = (deaths per million) / (seropositivity × 1000000) .

Several sentences are excessively long and would benefit from being split or simplified. Example: (lines 82–85): “Observation that most infections resulting in immunity were not reliably identified suggests that low virus exposures can sufficiently stimulate the immune system…”

Agreed.

Revised text:  Observation that most infections resulting in immunity were not reliably identified suggests a) that low virus exposures can sufficiently stimulate the immune system to stay ahead of the developing infection (in healthy people), b) that the exposure-response for immunity is stronger (requires less virus exposure) than that for developing serious clinical or fatal disease, and c) that number of new infections (conferring immunity) per fatality may be higher in populations with lower COVID-19 exposure levels.

The phrase “it will be argued here…” (line 48) is better suited to an opinion essay than a scientific article. Consider rephrasing as: “This analysis explores whether…”

Agreed.

Revised text:  A comparison of high COVID-19 mortality countries (e.g. U.S., U.K, Brazil, Italy) with low mortality countries (e.g. Taiwan, New Zealand, South Korea, Japan) during 2020 – 2024 provides an opportunity to explore the role of natural immunization in relation to the general levels of actual virus exposure over the course of pandemic mortality. This would require adopting fundamental concepts in a) environmental exposure assessment (surveys of airborne COVID-19 air concentrations) and b) epidemiological risk characterization (infection, case identification and fatality incidence rates) in relation to airborne virus exposure. The objective here is to describe a COVID-19 prevention strategy before and after vaccine availability, based on virus air-concentrations, seropositivity and mortality data, that would minimize the ultimate mortality burden and the associated social and economic disruption.

The assertion that Japan’s policy was “close to ideal” (line 285) should be framed more cautiously. This conclusion is based on retrospective modeling and assumptions about exposure levels that are not directly measured.

Agreed.

Revised text: These observations support the conclusion that an optimum COVID-19 control strategy would maintain a low but nonzero exposure level both before and after the advent of vaccination. The levels achieved by Japan (and possibly Vietnam) over the period 2020-2021 appear to have been in the optimum range.

In Section 3.3, the Chinese mortality estimate relies on cremation data (line 288), but no reference to how this data was validated is provided. Including citations for these methods and discussing potential biases would improve credibility.

I have replaced the validation statement to merely cite the agreement arising from independent sources.

Revised text: Making assumptions that infections rose relatively uniformly across many provinces, the national toll during the 2022-23 surge was estimated to be 1.5 million.36

...

The estimates for the surge from independent sources adds to the credibility of the 1.5 million cremation-based estimate.deaths

The manuscript could engage more deeply with recent literature on serological surveillance. For example, how does the proposed approach compare with established models for targeted serosurveys (e.g., in workplace settings or sentinel sites)?

Brief commentary on The Great Barrington and John Snow proclamations has been added (Section 4.2).

The focus of published serological surveillance studies has primarily been on point-of-care diagnostic practice. Two papers (Bonanni 2021, Barchuk 2022) have been cited (section 4.1)addressing serology in support of general pandemic epidemiological objectives and another (Winter 2022) discusses, in addition, the assignment of seropositive healthcare workers in high virus-exposure environments.

The discussion on mask efficacy and ambient virus exposure (lines 368–372) could be supported with citations from aerosol transmission studies or guidelines by WHO/CDC.

Line 36: “denial of pandemic” → revise to “pandemic denial” or “denial of the pandemic.”

It is difficult to evaluate studies of face-mask efficacy under typical-use conditions absent virus air concentrations. Moreover, I am not an authority on PPE or the current debate of mask efficacy. I am merely including masks among the possibly important determinants of virus transmission.

 “denial of pandemic” → revise to “pandemic denial”

Line 79: Consider rephrasing “CDC estimated that 60% had been infected” to clarify that this refers to estimated cumulative incidence, not active infection.

Agreed

Revised text:... had previously experienced a COVID-19 infection..

Table 2: Label units consistently (e.g., “Deaths per million” instead of just “Deaths/M”)

Agreed

This has been done throughout Tables and text.

Reviewer 3 Report

Comments and Suggestions for Authors

This interesting manuscript presents a comprehensive comparative analysis of COVID-19 pandemic outcomes across several countries with varied public health strategies, emphasizing the value of environmental exposure assessment and seropositivity surveillance. The analysis is data-rich, well-documented, and provides novel insights into alternative pandemic management approaches. The article offers valuable retrospective evaluations and thought-provoking recommendations for future pandemic preparedness.

The manuscript acknowledges limitations in data completeness, especially regarding seroprevalence and mortality attribution. However, the potential biases introduced by these limitations (e.g., underreporting in low-resource settings, differences in case definitions) should be more thoroughly discussed.

The use of ecological-level data to infer individual-level risk (e.g., exposure response) is inherently limited. This should be explicitly cautioned.

The concept of maintaining "low but nonzero virus levels" needs a more robust scientific grounding—are there precedents or theoretical models supporting this strategy?

Author Response

Reviewer 3

This interesting manuscript presents a comprehensive comparative analysis of COVID-19 pandemic outcomes across several countries with varied public health strategies, emphasizing the value of environmental exposure as

sessment and seropositivity surveillance. The analysis is data-rich, well-documented, and provides novel insights into alternative pandemic management approaches. The article offers valuable retrospective evaluations and thought-provoking recommendations for future pandemic preparedness.

The manuscript acknowledges limitations in data completeness, especially regarding seroprevalence and mortality attribution. However, the potential biases introduced by these limitations (e.g., underreporting in low-resource settings, differences in case definitions) should be more thoroughly discussed.

Low resource settings largely excluded in country choices

Case defns/mort attribs: mainly in early phase, sml impact on final cum mort

Revised text:  The study data-reporting limitations imply significant quantitative bias early in the pandemic but likely would have had a small impact on final cumulative mortality rates/million as increasing alignment on reporting protocols accompanied pandemic progress. These biases could not have accounted for much of the wide disparities observed between Western and East Asian societies.

The use of ecological-level data to infer individual-level risk (e.g., exposure response) is inherently limited. This should be explicitly cautioned.

Agreed.

Revised text: This study’s reliance on ecological-level data is a fundamental limitation, especially the implied uniformity of exposure within target populations, even entire countries. The study data-reporting limitations imply significant quantitative bias early in the pandemic but likely would have had a small impact on final cumulative mortality rates/million with increasing alignment of reporting protocols as the pandemic progressed. These biases could not have accounted for most of the wide disparities observed between Western and East Asian societies.  

The concept of maintaining "low but nonzero virus levels" needs a more robust scientific grounding—are there precedents or theoretical models supporting this strategy?

Probably not. Virus exposures appear not to have been systematically measured during past outbreaks and should be an important input in future pandemic modeling. This is a key recommendation posed in this work.

[407] Most environmental toxigens are regulated by way of a maximum allowed concentration, even carcinogens believed to have “no safe level.” Viral infections like COVID-19 and other infectious pandemic diseases are anomalous in that zero-exposure is considered an appropriate default objective even though lacking a scientific basis in risk assessment or public health strategy.

Round 2

Reviewer 2 Report

Comments and Suggestions for Authors

This paper provides a valuable, innovative perspective on COVID-19 mortality management strategies using global data and seropositivity surveys. The manuscript offers important insights into how alternative policies could have influenced mortality outcomes across countries.

To strengthen this paper, consider simplifying the methods section, which is currently dense with technical terms and layered explanations. Using subheadings or summarizing complex calculations in supplementary material would enhance readability. Similarly, while your results are detailed and robust, highlighting key findings at the start of each results subsection would help guide readers through your narrative. Improving the clarity of figures, including enlarging labels and simplifying plots where possible, would further enhance accessibility.

Your conclusions are well-supported by your results and contribute meaningfully to discussions on pandemic policy and preparedness. With these refinements in clarity and presentation, your important findings will be more accessible to both policymakers and the broader public health research community.

Comments on the Quality of English Language

The language in the manuscript is generally clear and understandable, allowing the scientific content to be followed. However, there are areas where the language could be improved to enhance clarity and flow. Some sentences are long and complex, making it difficult for readers to quickly grasp key points. Shortening these sentences and using more direct phrasing would improve readability. Additionally, there are minor grammatical errors and occasional awkward phrasing that should be corrected during revision. Overall, with careful language editing, the clarity and impact of the paper will be significantly improved.

Author Response

This paper provides a valuable, innovative perspective on COVID-19 mortality management strategies using global data and seropositivity surveys. The manuscript offers important insights into how alternative policies could have influenced mortality outcomes across countries.

To strengthen this paper, consider simplifying the methods section, which is currently dense with technical terms and layered explanations. Using subheadings or summarizing complex calculations in supplementary material would enhance readability.

Multiple subheadings with brief introductory text have been added in Methods, Results and Discussion (see revision comparison file).

Methods has been re-organized to consolidate and simplify (p. 6, 7).

The following text was added on the prediction algorithm method:

The proportion at risk of infection (unvaccinated and no prior infection, PaR), needed for predicting the final burden, can be estimated as follows:

PaR = 1-(SP × (1-%VAC/100) × 0.90 + (1-SP) × %VAC/100 × 0.95 + SP × %VAC/100 × 0.98)

where: SP is natural COVID-19 seropositivity; SP = cumulative mortality per million13 × FIR /1,000,000; %VAC = percent  completing full original vaccination protocol13; fatality protection factor (PF) with prior infection is stipulated to be 90%, with vaccination is 95% and with both vaccination and prior infection is 98%. This expression is one minus the proportion protected, and the protected are a sum representing three distinct subpopulations: prior infected/no vaccination, no prior infection/vaccinated, and prior infection/vaccinated. This calculation assumes that the FIR, even though measured in most cases prior to vaccination, applies in the future. At the end of 2024 the PaR was specified to be:

PaR(1 Jan 2025) = 1-(0.95×0.05×0.90 + 0.05×0.95×0.95 + 0.95×0.95×0.98)

where the final SP and %VAC were assumed to equal 95%. The COVID-19 mortality rate in the unprotected population (TMR) was calculated from the observed overall mortality rate (OMR) at 1 Jan 2023 as follows:  TMR = OMR/PaR(1 Jan 2023). In predicting the additional COVID-19 mortality during 2023-2024 the rate TMR was applied to the mean of PaR(01-01-2023) and PaR(01-01-2025):

deaths (2023-2024) = population × 2 yr × TMR × (PaR(01-01-2023) + PaR(01-01-2025))/2 .

Similarly, while your results are detailed and robust, highlighting key findings at the start of each results subsection would help guide readers through your narrative. Improving the clarity of figures, including enlarging labels and simplifying plots where possible, would further enhance accessibility

Headings added and introduced.

Fig.2 has been enlarged and labels added and enlarged. The legend has been expanded: Cumulative confirmed Covid-19 deaths per million for East Asian countries and New Zealand, which exhibited low death rates...

Producing new versions of the Fig 1 & 2 plots is no longer supported by the graphics interface used earlier.

Added txt:

3.2       Seropositivity and COVID-19 exposure intensity

In the countries and regions with concurrent seropositivity survey and cumulative fatality data, the evidence supports the conjecture that the FIR decreases with increasing average COVID-19 mortality rate (surrogate for COVID-19 exposure). In the Western countries with high COVID-19 mortality the mean relative exposure measure was 119 with corresponding geometric mean FIR of 179 (Supplementary file, Table S5). The East Asian and New Zealand countries had a mean relative exposure measure of 0.47 with corresponding geometric mean FIR of 1466, but some seropositivity measures were very early in the pandemic and probably unrepresentative. Plots of FIR with relative exposure reveal a trend of decreasing FIR by an order of magnitude or more over the range of increasing relative exposure (Supplementary file, Figs. S1, S2), but the data are limited.

New subheadings:

3.4       Patterns of mortality rates in response to apparent changes in COVID-19 exposure

3.5       Effect on natural immunity of prolonging zero-COVID controls                                  

Comments on the Quality of English Language

The language in the manuscript is generally clear and understandable, allowing the scientific content to be followed. However, there are areas where the language could be improved to enhance clarity and flow. Some sentences are long and complex, making it difficult for readers to quickly grasp key points. Shortening these sentences and using more direct phrasing would improve readability. Additionally, there are minor grammatical errors and occasional awkward phrasing that should be corrected during revision. Overall, with careful language editing, the clarity and impact of the paper will be significantly improved.

Simplified text:

p.6  One study bearing on the infection exposure response examined seropositivity in the 2,474,066 contacts traced from 1,064,004 presenting cases in England during 1 Sept 2020 to 28 Feb 2021.11 231,498 of contacts (9%) had COVID-19 PCR-seropositivity, the prevalence of which increased almost 4-fold across 5 levels of viral load (PCR assay) in the presenting case (p<0.001).

Numerous minor edits were needed and have been made to improve clarity and flow. These are all identified in revision comparison file.

The author greatly appreciates the reviewer’s input and considerable efforts toward these improvements.

Your conclusions are well supported by your results and contribute meaningfully to discussions on pandemic policy and preparedness. With these refinements in clarity and presentation, your important findings will be more accessible to both policymakers and the broader public health research community.

Submission Date        29 April 2025

Date of this review    03 Jul 2025 21:01:58